# Epigenetic Modulation and Neuroprotective Effects of Neurofabine-C in a Transgenic Model of Alzheimer’s Disease

**DOI:** 10.3390/genes16101214

**Published:** 2025-10-15

**Authors:** Ivan Carrera, Vinogran Naidoo, Lola Corzo, Olaia Martínez-Iglesias, Ramón Cacabelos

**Affiliations:** EuroEspes Biomedical Research Center, International Center of Neuroscience and Genomic Medicine, 15165 Bergondo, Corunna, Spain; neurociencias@euroespes.com (V.N.); analisis@euroespes.com (L.C.); epigenetica@euroespes.com (O.M.-I.); rcacabelos@euroespes.com (R.C.)

**Keywords:** neuroinflammation, cocoa, neurodegeneration, animal model, neurogenesis

## Abstract

**Background**: Currently, there are limited therapeutic or preventative strategies for neurodegenerative disorders due to the challenges in alleviating the progressive neuronal loss and neuroinflammation which are the primary characteristics of these diseases, ultimately leading to cell death and functional impairment. Cocoa-derived flavanols (*Theobroma cacao*) have been studied as potential bioactive compounds to modify and reverse various inflammation-associated diseases because of their remarkable antioxidant properties and capacity to modulate metabolic imbalance and reactive inflammatory responses. The faba bean (*Vicia faba*) extract obtained through nondenaturing biotechnological processes is a potent dopamine (DA) enhancer that has shown promising results as a neuroprotective agent against degeneration. **Objective**: This study will examine the synergistic effects of Neurofabine-C, a hybrid compound derived from cocoa and faba bean extracts, on various brain biomarkers in mice related to inflammatory, metabolic, and neurodegenerative processes. **Methods**: A triple-transgenic mouse model of neurodegeneration was treated with Neurofabine-C, and biomolecular data were obtained by performing biochemical and immunohistochemical analysis. **Results**: Neurofabine-C prevented neuronal degeneration (NeuN), mitigated the neuro-inflammatory processes triggered (decreased expression of reactive astrocytes (GFAP)), and induced an increase in neurogenesis in the treated cortical mice brain (PAX6). Epigenetic analysis revealed significant chromatin remodeling in the hippocampus. Neuroprotective genes, including *FOXO3*, *ATM*, and *TRP73*, were upregulated, whereas the expression of *HIF1α* and *APOE* decreased. In parallel, *DNMT3A* expression increased 20-fold, *HDAC3* decreased by 60%, and global 5-methylcytosine levels increased four-fold. These coordinated changes suggest that Neurofabine-C promotes neuroprotective programs through enhanced DNA methylation and reduced histone deacetylation. **Conclusions**: The findings indicate that Neurofabine-C exhibits multiple neuroprotective mechanisms, making it a potent bioproduct for mitigating neuroinflammatory processes associated with neurodegenerative disorders.

## 1. Introduction

Neurodegenerative diseases represent a broad group of disorders characterized by the progressive loss of neuronal structure and function in various regions of the brain or spinal cord. Examples include Parkinson’s disease (PD), Alzheimer’s disease (AD), Huntington’s disease, multiple sclerosis, amyotrophic lateral sclerosis and Creutzfeldt–Jakob disease [1]. Although each disease manifests with unique clinical features and pathological profiles, they are unified by the gradual degeneration of nerve cells. The origins of these conditions are complex, involving an interplay of genetic susceptibility, environmental exposures, age-related processes, misfolded protein deposits, and other pathological alterations within the brain. Currently, no curative treatments exist; however, available neuroprotective strategies can slow disease progression and improve the quality of life for affected patients.

Neuroprotection refers to the ability of bioactive molecules or endogenous factors to prevent neuronal damage or death by maintaining the integrity and proper functioning of nerve cells. This protective effect is primarily mediated through neurotrophic activity, which supports neuronal growth, differentiation, and activity while triggering multiple defense mechanisms. These include reducing oxidative stress, suppressing inflammation, regulating programmed cell death, enhancing mitochondrial function, and improving cerebral blood flow. Key neurotrophic molecules such as nerve growth factor (NGF), brain-derived neurotrophic factor (BDNF), and glial-cell-line-derived neurotrophic factor (GDNF) are fundamental for the formation, maintenance, and repair of the nervous system. By binding to receptors on neurons, they regulate survival, synaptic activity, and neural plasticity. Therefore, compounds that mimic these factors or stimulate their production hold potential as therapeutic agents for neurodegenerative diseases [2].

The use of natural product mixtures or extracts containing multiple bioactive compounds in neuroprotection is a focus of rigorous scientific inquiry designed to ascertain any health advantages and elucidate the mechanisms of their action at the neural level. Research suggests that several natural bioactive chemicals exhibit potential neuroprotective effects, creating opportunities for their application in the prevention and treatment of neurological disorders [3,4]. Among them, flavanols produced from cocoa are being studied as possible bioactive compounds to modulate and reverse various inflammation-associated diseases because of their remarkable antioxidant qualities and capacity to modify metabolism, but mainly due to the modulation against a reactive inflammation response.

It is widely known that polyphenols are abundant in cocoa, tea, coffee, fruits, and vegetables. Low-molecular-weight flavanols, like epicatechin, are thought to be particularly important in this complex group of chemicals because of their antioxidative qualities, which have drawn a lot of attention and may have positive impacts on human health [5,6]. The unroasted beans of *Theobroma cacao* L., an evergreen cacao tree, are known as cocoa. Particularly well-known are its effects on behavior regulation, which primarily impact mood, appetite, and relaxation. However, numerous studies have documented the advantages of cocoa extracts and their components in reducing inflammation or immune system impairment, aging, controlling blood pressure, preventing atherosclerosis, and preventing the development of cardiovascular disorders [7,8,9]. Platelet activation [10,11], nitric oxide (NO)-dependent activities [12,13,14], blood pressure [15], insulin resistance [16], and cytokine production [17] have all been demonstrated to be impacted by cocoa components. Although its effectiveness is dependent on the level of the basal inflammatory burden, there is some convincing evidence that eating foods high in cocoa may reduce inflammation, mostly by reducing the activation of monocytes and neutrophils [18]. As a result, a primary focus of preventative measures should be the inflammatory background in disease development, which includes the initiation of pro-inflammatory cytokine release and metabolite synthesis. Thus, by concentrating on pathways involved in cell-mediated immune response in the brains of animal models, we hope to address possible interferences of cocoa antioxidants with central immunoregulatory processes. Additionally, interferences with vascular function, inflammatory processes and platelet reactivity are among the most talked-about effects of cocoa absorption, indicating that certain elements may have vascular-protective qualities [19,20]. Lipid accumulation in the arterial wall causes chronic inflammation, which thickens and hardens the vessel wall, decreasing its flexibility and obstructing blood flow [21]. Consuming a diet rich in antioxidants may reduce the likelihood of developing atherosclerosis by limiting oxidative reactions and modulating cellular signaling pathways [22,23].

Recent studies have demonstrated that phytochemicals present in fruits, vegetables, herbs, spices, and algae exhibit complementary antioxidant and anti-inflammatory properties, which may contribute to the health-promoting effects of plant-based diets [23]. One notable example is E-PodoFavalin-15999 (AtreMorine), a novel extract obtained from the structural components of *Vicia faba* L. using non-denaturing biotechnological techniques [24,25,26,27]. This preparation is a natural source of L-DOPA and acts as a potent stimulator of plasma catecholamines (noradrenaline, adrenaline, and dopamine) while leaving serotonin concentrations unaffected [28]. In addition, it provides a wide range of bioactive constituents, including plant proteins, unsaturated fatty acids, vitamins, minerals, dietary fiber, starch, pigments such as carotenes, and phytosterols. Phytoconstituents derived from *Vicia faba* L. have shown distinct antioxidant and neuroprotective activities, as confirmed through in vitro studies, animal experiments, and clinical investigations.

In vitro studies have shown that AtreMorine is a strong neuroprotectant in several experimental models, including (i) human SH-SY5Y neuroblastoma cell cultures, (ii) hippocampal slices subjected to oxygen and glucose deprivation, and (iii) striatal slices exposed to 6-OHDA-induced toxicity [24]. Across these systems, AtreMorine demonstrated both neuroprotective and anti-inflammatory properties, suggesting its potential value in protecting neurons from degeneration, particularly dopaminergic cells implicated in the development of PD. Furthermore, in vivo experiments revealed that AtreMorine (i) prevents dopaminergic neurodegeneration caused by 1-methyl-4-phenyl-1,2,3,6-tetrahydropyridine (MPTP), (ii) reduces microglial activation and neurotoxicity in the substantia nigra induced by MPTP, and (iii) improves motor performance in mice with MPTP-related neurodegeneration [25].

AtreMorine elevates plasma dopamine concentrations in patients with PD and related movement disorders by approximately 200–500-fold, reflecting its strong capacity to enhance dopaminergic neurotransmission. This effect occurs not only in drug-naïve individuals but also in those undergoing long-term therapy with L-DOPA or other antiparkinsonian agents [27,28,29]. Conventional treatments are frequently associated with adverse effects, including the “wearing-off” phenomenon, motor fluctuations, dyskinesias, systemic complications such as gastrointestinal or cardiovascular disturbances, endocrine imbalances, and neuropsychiatric symptoms, including depression, anxiety, or toxic psychosis. AtreMorine has demonstrated the ability to mitigate many of these complications. When administered in combination with standard antiparkinsonian drugs, it allows for a 25–50% reduction in dosage while simultaneously enhancing therapeutic efficacy, minimizing both short- and long-term side effects, and extending the duration of clinical benefit.

To stop or mitigate the effect of neuroinflammation and cognitive decline indicators in AD transgenic mouse models, our guiding framework focused on Neurofabine-C, a hybrid compound integrating cocoa and faba bean extract. As a result, we postulated that a brief treatment with Neurofabine-C may lower the levels of important neuroinflammatory and degenerative indicators, which would then affect brain protection. By examining specific mouse brain markers of inflammatory, metabolic, and neurodegenerative processes, the overarching goal of this work is to gain a deeper understanding of the protective benefits of Neurofabine-C-enriched extract against neurodegenerative diseases.

## 2. Materials and Methods

### 2.1. Mouse Models

The Swedish mutation of APP (human amyloid precursor protein), BIN1 (bridging integrator 1, AMPH2), and COPS5 (COP9 constitutive photomorphogenic homolog subunit 5, Jab1) is overexpressed in triple-transgenic 3xTg-AD mice (APP/BIN1/COPS5). Dr. Laksmana’s lab graciously contributed mice, which we then backcrossed in our lab. With the consent of the EuroEspes Research Center’s Ethics Committee (Permit number: EE/2015-332), all experimental procedures were carried out in accordance with Spanish law (R.D. 1201/2005), EU Directive 2016/63/EU, and European Community law (86/609/EEC).

Twelve-week-old AD transgenic mice (APP/BIN1/COPS5; 3xTg-AD) were treated with commercial feed ad libitum or feed prepared with cocoa-bean extract ad libitum (Table 1). For this study, five mice were used in the control group A (without treatment in food) and 10 in treatment group B (with Neurofabine-C enriched food). Mice were kept in an environment with regulated humidity (40–50%), temperature (20–21 °C), and lighting (12 h of light/dark cycle). They were also given unlimited access to water. After five weeks of treatment, a total of fifteen mice were killed, and samples of their blood and brain were taken.

All experimental mice procedures were conformed to the guidelines established by the European Communities Council Directive (86/609/EEC), the EU Directive 2010/63/EU, and the Spanish Royal Decree 1201/2005 for animal experimentation and were approved by the Ethical Committee of the EuroEspes Biotechnology Research Centre (Permit number: EE/2023-08).

The table shows the experimental protocol, displaying mice in groups A and B fed a conventional diet (RD) and a diet supplemented with cocoa-fava bean extract (CES) over a 5-week period.

### 2.2. Biochemical Characterization of Neurofabine-C Extract

Neurofabine-C, a hybrid compound, was obtained from combining high-nutritional-value bioproducts such as the fava bean (*Vicia faba*) with cacao powder (*T. cacao*). The organic cacao powder was made from hand-harvested raw cacao beans in Peru, fermented to lessen bitterness, and then extracted to be broken and ground into cacao paste, preserving all of the cacao’s nutritional value, according to the supplier (Bulk Powder Co., batch:023-2245, Colchester, England). The final combined extract was produced through non-denaturing biotechnological techniques, including lyophilization under regulated temperature, pressure, and grinding conditions (Patent ID: P202230047/ES2547.5). Refer to the Appendix A for further information on nutrition analysis. Treatment preparation: Using diet wheat as the primary flour, 50% powder treatment (70% cocoa, 30% fava bean in 100 g of final bulk) was added to the diet as pellet biscuits. 10% (*w*/*w*) MilliQ-purified water was added for pelleting, and the pellets were then dried at 34 °C for the entire night. The main nutritional composition of the treatment diet is shown in the Appendix A.

### 2.3. Blood Sample Preparation

Following deep anesthesia of the mice, a heart puncture was used to collect blood at the conclusion of the experiment. Commercial tubes of the Greiner Bio-One brand were utilized; some had EDTA K3 as an anticoagulant, while others did not. Blood cells (LEU, RBC, PLQ), hemoglobin (HGB), hematocrit (HTO), and erythrocyte indices were counted in the intact, unprocessed EDTA tube using the MINDRAY BC-5380 hematological analyzer. After the blood counts, the plasma was extracted from the EDTA tubes by centrifuging them for 10 min at 4000 rpm and 4 °C. The plasma was then stored at −80 °C until the vitamin B9 (folate) test. Vitamin B9 (folate) was measured using an Elabscience (Houston, TX, USA) commercial ELISA kit and an absorbance measurement at 450 nm in the EPOCH reader (Biotek Instruments, Winooski, VT, USA). After extraction, the anticoagulant-free tubes were left to clot upright for 20 min before being centrifuged for 10 min at 4000 rpm to collect the serum from the cell layer. The serum was stored at −80 °C until analysis. Total antioxidant status rate (TAS) in serum samples was measured using Randox total antioxidant status kit (Randox Laboratories Ltd., Crumlin, United Kingdom) and albumin using Biosystems reagents (Barcelona, Spain) adapted to the COBAS MIRA automated chemistry analyzer (Roche Diagnostic Systems, Mannheim, Germany) through UV–visible spectrophotometry using a Cobas Mira Plus analyzer (Roche Diagnostics, Basel, Switzerland) and vitamins B6, B12, and β-amyloid 1-42 (βA42) using commercial ELISA kits from Elabscience (USA), DRG International (USA), and Elabscience (USA), respectively. The protocols recommended in the techniques manual were followed in all cases. The reactions were read according to absorbance at 450 nm using an EPOCH reader (Biotek Instruments, USA).

### 2.4. Brain Sample Preparation

Anesthetized animals underwent transcardial perfusion with 0.9% NaCl followed by 4% paraformaldehyde (PFA; Cat. #43368, Alfa Aesar™, Germany). Mouse brains were removed and post-fixed in 4% PFA for 48 h following perfusion. The tissue was then cryoprotected in 30% sucrose made in PB, embedded in OCT compound (Tissue-Tek, Torrance, CA, USA), and frozen using isopentane cooled with liquid nitrogen after being moved to 0.1 M phosphate buffer (PB, pH 7.4) for 12 h. The right hemisphere was sectioned on a cryostat into serial transverse slices (18 μm thick), while the left hemisphere was dissected to isolate the hippocampus for gene expression analysis. The cryosections were placed on Superfrost Plus slides (Menzel Glasser, Madison, WI, USA) and kept at room temperature for histological processing. For the determination of brain βA42, two areas were chosen, neocortex and hippocampus, as they are considered the most associated with cognitive impairment. The samples were processed according to the recommendations indicated in the technique protocol for the preparation of a tissue homogenate. The different brain parts were extracted from the animal and cut into small pieces and rinsed in ice-cold PBS (0.01 M, pH 7.4) to remove excess blood and avoid interference in subsequent readings. The tissue pieces used in the analysis were weighed and then homogenized in PBS in a 1:9 ratio (tissue weight/PBS volume) using a glass homogenizer on ice. For further cell breakdown, the suspension was sonicated with an ultrasonic cell disruptor. The homogenates were centrifuged for 5 min at 5000× *g* to obtain the supernatant, which was frozen at −80 °C until analysis.

### 2.5. Epigenetic Expression Analysis

After five weeks of treatment, hippocampus samples were collected and the expression of MRPE corresponding to different genes involved in neurodegeneration and neuroprotection was analyzed, as well as in the epigenetic regulation of gene expression. For global methylation analysis, a colorimetry kit (Abnova) was used.

**RNA Extraction:** Total RNA from peripheral blood lymphocytes was extracted using the PureLinkTM RNA Mini Kit (Invitrogen, Waltham, MA, USA). The RNAEasy Mini Kit (Qiagen) was used to extract total RNA from the livers of mice. In short, samples were treated with lysis buffer and 2-mercaptoethanol after being centrifuged to eliminate the Qiazol reagent. After that, the lysates were moved to purification columns and exposed to Invitrogen’s Pure-LinkTM DNAse. Following a series of washing procedures, RNA was eluted using water free of RNAse, and its concentration and purity were assessed. In this investigation, only RNA samples with 260/280 and 260/230 ratios higher than 1.8 were employed.

**DNA Extraction:** Genomic DNA was isolated from mouse hippocampal tissue using the Qiagen DNA Mini Kit (Qiagen, Hilden, Germany) according to the manufacturer’s protocol. Only samples with 260/280 and 260/230 absorbance ratios above 1.8 were included for downstream analysis.

**Quantification of Global DNA Methylation (5 mC):** Overall 5-mC levels were determined with a colorimetric ELISA-based methylated DNA quantification kit (EpigenTek, New York, NY, USA). For each assay, 100 ng of DNA was used. Following the recommended sequence of reactions and washes, absorbance was recorded at 450 nm. Relative quantification was performed as instructed by the manufacturer, using the formula5-mC = ((Sample OD − Negative OD)/DNA (ng))/((Positive OD − Negative OD) × 2/amount positive control (ng)).

**Quantitative Real Time RT-PCR:** The High Capacity cDNA Reverse Transcription Kit (Applied Biosystems, Waltham, MA, USA) was used to reverse-transcribe the RNA in accordance with its instructions. For the retrotranscription reaction, 400 ng of RNA was employed, and the thermocycling conditions were 10 min at 25 °C, 120 min at 37 °C, and 5 min at 85 °C.

The expression of *APOE*, *FOXO3*, *ATM*, *HIF1α*, *TRP73*, *HDAC3*, and *DNMT3a* was measured through quantitative PCR (qPCR) on a StepOne Plus Real-Time PCR system (Applied Biosystems) according to the manufacturer’s protocol. Reactions were carried out in duplicate using TaqMan Gene Expression Master Mix (NZYTech) together with TaqMan probes (Thermo Fisher, Waltham, MA, USA). Relative transcript levels were calculated using the comparative CT method with StepOne Plus software. Human glyceraldehyde-3-phosphate dehydrogenase (GAPDH) served as the internal control for normalization. Results are expressed as fold changes relative to healthy controls and presented as mean ± S.E.M.

### 2.6. Immunohistochemistry

Immunohistochemical characteristics were examined in accordance with other publications [30]. To summarize, slices of mouse brain were incubated at 4 °C for one entire night with primary antibodies against the neuronal nuclear protein NeuN (1:1000; MAB-377) (Millipore, Madrid, Spain), Pax-6 neuronal protein (1:1000; 42-6600, Invitrogen, Madrid, Spain), and glial fibrillary acidic protein GFAP (1:800; MA5-12023, Invitrogen). The Alexa Fluor-tagged secondary antibody (Thermo Fisher Scientific, Madrid, Spain) was used to detect the neuronal proteins. By leaving out the primary antibody, the fluorescent immunostaining’s specificity for each antibody was verified. The slices were then dyed using DAPI (Vector Laboratories, Newark, CA, USA).

### 2.7. Imaging

A Leica DM6-B upright microscope (Leica Microsystems, Buffalo Grove, IL, USA) and Leica Application Suite X (LAS X) software were used to record fluorescence signals. The mean density of the triplicates of immunofluorescence cell markers relative to the background in each brain segment image was calculated using the area/pixel analysis program (Pixcavator 4).

### 2.8. Statistical Analysis

The statistical program SPSS (version 23.0) was used to analyze the data. To ascertain whether the sample was parametric or not, a test for homogeneity of variance (Levene’s test) was conducted first. Using this criterion, the *t*-test for related samples was used to compare the means of the two groups (extract group and control group) with their standard deviations/errors. The determination of statistical significance was made when *p* was 0.05 or less.

## 3. Results

### 3.1. Biochemical Effects of the Neurofabine-C on Metabolic and Neurodegeneration Response

Table 2 presents the data of all the analytical parameters performed on this group in order to see the effectiveness of the nutritional treatment provided according to protocol on the nutritional (albumin and vitamins), antioxidant (TAS and MDA) and cognitive status associated with Alzheimer’s-type dementia (βA42).

Table 2 analyzed parameters measured in the experimental groups to address the effectiveness of Neurofabine-C on the biochemical state based on mice metabolism and neurodegeneration markers. *p* value of 0.05 or less is considered statistically significant. ALB: Albumin; MCH: Mean Corpuscular Hemoglobin; MCHC: Mean Corpuscular Hemoglobin Concentration; MCV: Mean Corpuscular Volume; MDA: Malondialdehyde; MPV: Mean Platelet Volume; PCT: Platelet crit; TAS: Total antioxidant status.

Regarding the cellular immune system (WBC) and the traditional indicators of anemia and energy metabolism (HGB, RBC, and HCT), no discernible alterations were seen between the control and treated groups.

In the present study, changes in antioxidant/oxidant factors were assessed by measuring plasma levels of TAS and MDA. TAS, a measure of overall antioxidant capacity, describes the dynamic equilibrium between different prooxidants and antioxidants in blood [31]. In this animal model of Alzheimer’s-type neurodegeneration, a significant increase in TAS capacity compared to the control group reflects the important antioxidant and, consequently, neuroprotective effect of Neurofabine-C. However, we found a non-statistically significant upward trend in MDA levels after treatment, which do not favor its antioxidant power (Figure 1A,B).

MDA and TAS levels were analyzed in brain tissue (neocortex and hippocampus). Many of the values in both areas of the mouse brain have fallen below the detection limits of the kits, so we have discarded these results in order to make a correct assessment. Albumin, a recognized biomarker used in most nutritional studies, was used to monitor the nutritional value of Neurofabine-C. No significant changes were found in the group of mice that received Neurofabine-C, although a tendency towards increased plasma levels was observed (Figure 1C).

B vitamins, particularly B6, B9, and B12, have been shown in numerous trials to have positive benefits in preventing and enhancing cognitive decline. However, we did not observe significant differences in the plasma levels of vitamins B6, B9 and B12 analyzed in our study after Neurofabine-C intake (Figure 2).

In order to address the neurodegenerative impact in the AD brain of mice, β-Amyloid 1-42 (βA42) protein levels were examined because of their correlation with cerebral amyloidosis and cognitive deterioration leading to dementia in patients, and they are presently regarded as the most specific biomarker for Alzheimer’s dementia.

In our basic study in mice (βA neurodegeneration model), we found that βA42 levels slightly increased in serum and significantly decreased in the neocortex, with no differences found in the hippocampus (Figure 3).

### 3.2. Epigenetic Modulation of Neurodegeneration-Related Pathways

#### 3.2.1. Neurodegenerative-Related Gene Expression in the Hippocampus of AD Mice

Deficits in DNA repair contribute to the accumulation of chromosomal damage in neurons, leading to dysfunction and neurodegeneration. In order to address the epigenetic effect of Neurofabine-C, crucial genes were analyzed such as Forkhead Box O3 (FOXO3) that plays a central role in neuroprotective pathways, ATM (ataxia telangiectasia, mutated), another key DNA repair regulator and HIF-1α, elevated expression of which is linked to neurodegenerative diseases, inflammatory processes, and post-stroke pathology. Present results show that Neurofabine-C significantly modulated the expression of these genes in AD transgenic mice: FOXO3 expression increased 2.5-fold, ATM expression increased 14-fold, and HIF-1α expression decreased 3-fold, compared to controls (Figure 4A).

In addition to FOXO3 and ATM, TRP73 is directly involved in neuronal differentiation and plays a key role in the DNA damage response. Transgenic animals lacking *TRP73* exhibit neurological and immunological defects. In our study, treatment of AD transgenic mice with Neurofabine-C induced a more than 20-fold increase in *TP73* expression compared to control animals (Figure 4B).

APOE is strongly implicated in neurodegeneration and in the early development of key AD hallmarks. Increased *APOE* expression has been reported in conditions of brain damage and in the hippocampus of patients with AD. In our study, treatment of AD transgenic mice with Neurofabine-C reduced *APOE* expression by approximately 40% compared to controls (Figure 4C).

Finally, we examined the expression of *HDAC3* and *DNMT3A* in the hippocampus of AD transgenic mice and control animals. In this model, treatment with Neurofabine-C markedly increased *DNMT3A* expression approximately 20-fold in 3xTg-AD mice. DNMT3A is a key DNA methyltransferase that regulates activity-dependent gene expression, and its upregulation suggests increased capacity for transcriptional regulation of neuronal survival and memory-related genes. Since the balance between histone acetylation and deacetylation is crucial for neuronal homeostasis, excessive deacetylation has been strongly associated with neurodegenerative processes. In line with this, Neurofabine-C reduced *HDAC3* expression by approximately 60%. Given the role of HDAC3 as a negative regulator of memory consolidation, its downregulation may promote histone acetylation in hippocampal neurons, thereby counteracting epigenetic repression and mitigating neurodegenerative mechanisms in AD (Figure 4D).

#### 3.2.2. Effect of Neurofabine-C on Global Methylation Levels

Global levels of 5-methylcytosine (5mC) have been proposed as a critical biomarker in AD, since patients with neurodegenerative disorders, including AD, typically display reduced DNA methylation. To assess whether cocoa-bean extract influences this epigenetic marker, we measured hippocampal 5mC levels in 3xTg-AD and control mice. Treatment with Neurofabine-C resulted in an approximately four-fold increase in global methylation in transgenic mice. This effect is consistent with the marked upregulation of *DNMT3A* observed in treated animals, suggesting increased DNA methyltransferase activity. The elevation of 5mC levels in hippocampal tissue indicates that Neurofabine-C supports epigenetic stability, providing a potential anti-degenerative mechanism in the AD brain (Figure 4E).

### 3.3. Preventive Effect of Neurofabine-C on Neurodegeneration in AD Mice

To investigate the preventive neurodegeneration effect of Neurofabine-C supplementation diet on the AD transgenic mice model, we performed an immunofluorescence staining analysis using antibodies against Pax6, as a key marker for differentiation of neural stem cells (Figure 5A,B), and NeuN as a marker for neuronal development.

In order to observe the different neuroprotective patterns between the experimental AD transgenic mice groups, details of the affected brain regions were analyzed (Figure 6). The cortical and hippocampal layers of the treated mice exhibited a pronounced distribution of immunoreactive Pax6 cells, predominantly concentrated in the outer layers of the cortex (Figure 6A’,B’), in stark contrast to the sparse density of immunoreactivity found across the entire region of the control mice. The entorhinal cortex of the treated mice showed an intense distribution of immunoreactive NeuN cells, mainly gathered in the outer layers of the cortex (Figure 6A’,B’), although a profuse distribution of this marker was observed throughout the entire region. In contrast, the same cortical region of the control 3xTg mice showed a scarce presence of immunoreactivity against NeuN, with only a particular cluster or accumulation site within the analyzed region (Figure 6A’,B’).

## 4. Discussion

The peels of fruits and vegetables are frequently thrown away or, at most, used as cow feed or fertilizer, which has serious negative effects on the environment [32]. There has been a recent reversal in the direction of reevaluating and valuing agro-food waste, which has been encouraged by a number of scientific research works. In fact, numerous studies have documented a high concentration of bioactive chemicals in some by-products, which are crucial as nutraceuticals with numerous health benefits, such as anti-oxidant and anti-cancer properties [33,34,35,36]. Numerous studies have documented the high nutritional value of legumes, including secondary metabolites like polyphenols, whose biological characteristics have been extensively researched [37]. These compounds, in particular, have well-known antioxidant qualities because they can counteract reactive oxygen and nitrogen species that are created as byproducts of metabolic processes [38]. Furthermore, a number of studies have shown that polyphenols offer a strong defense against chronic illnesses like cancer, diabetes, heart disease, infections, aging, and asthma [39]. Moreover, many recent studies have focused on the nutritional and bioactive properties of faba beans [40,41,42,43,44]. Beyond its nutritional value, the faba bean is a rich source of bioactive chemicals that have been shown to have positive health effects. These consist primarily of bioactive peptides [40,41] but also contain phenolic substances [30], resistant starch [45], dietary fibers [46], and non-protein amino acids (L-DOPA [47], GABA [48]). The nutritional properties (amino acid profile and digestibility), health-promoting bioactivities, and other matrix elements that may have both positive and negative nutritional and bioactive impacts are some of the factors that affect the quality of faba bean protein. Most faba bean proteins are globulin-types, which is common to most pulses [49]. Plant proteins are divided into four major classes according to their solubility in different solvents: globulins dissolve in dilute salt solutions, albumins in water, prolamins in 70% ethanol, and glutelins in alkaline buffers [50]. In faba beans, globulins represent the predominant fraction, accounting for 69.5–78.1% of total seed protein, followed by glutelins (12.0–18.4%), prolamins (1.83–3.57%), and albumins (1.41–3.01%) [51]. The relative abundance of these fractions varies with cultivar type and environmental conditions [50]. Within the globulin family, two principal subgroups are distinguished by their sedimentation coefficients: legumins (11S) and vicilins (7S) [52]. Up to 55% of the seed proteins in faba beans are legumins, the most prevalent globulins [53]. Here, we suggest valuing the synergic combination of cocoa and faba bean extracts (Neurofabine-C) as a source of antidegenerative and antioxidant biochemical modulators, examining these qualities using three distinct biological approaches: immunohistochemistry, biochemical, and epigenetic study.

Antioxidant intake has been associated with reduced oxidative damage to lymphocyte DNA. Comparable results have also been observed for foods and beverages rich in polyphenols, suggesting a protective role for these compounds [54]. Increasing evidence suggests that polyphenols, through their antioxidant activity, can protect cellular components against oxidative stress and thereby reduce the risk of various degenerative diseases associated with this process [55,56]. Regular consumption of fruit and vegetable juices containing high polyphenol levels, at least three times per week, has been reported to delay the onset of AD [57]. Due to their influence on multiple cellular functions, such as signaling, proliferation, apoptosis, redox regulation, and differentiation, dietary polyphenols are considered promising candidates for neuroprotection [58]. In line with this, a recent study [59] demonstrated that polyphenol administration protects against PD, a neurodegenerative disorder characterized by the loss of dopaminergic neurons in the substantia nigra pars compacta. Nutritional studies have also linked green tea intake with a reduced risk of developing PD. Experimental research in animal models has shown that epigallocatechin gallate (EGCG), a major green tea catechin, protects against MPTP (N-methyl-4-phenyl-1,2,3,6-tetrahydropyridine)-induced Parkinsonian neurodegeneration, either by scavenging radicals formed during MPTP metabolism or by competitively inhibiting toxin uptake due to its structural similarity. EGCG also promotes neuronal survival by activating multiple signaling cascades, including MAP kinases [60]. In addition, catechins contribute to neuroprotection through their iron-chelating activity, which prevents redox-active metals from catalyzing the formation of free radicals. Their antioxidant capacity is further reinforced by the induction of antioxidant and detoxifying enzymes, a mechanism fundamental in the brain, where endogenous antioxidant defenses are relatively weak [59].

Evidence points to the oxidation of biological substrates (ROS) by free radicals as a major aspect of AD pathogenesis. More recent research has shown elevated lipid peroxidation (LPO) biomarkers in the postmortem brain of subjects with mild cognitive impairment, the earliest clinically detectable phase of dementia, and preclinical AD, the earliest detectable pathological phase, although it has long been known that these biomarkers are elevated in the AD brain along with postmortem ventricular fibrillation. Additionally, as the disease progresses, a number of LPO indicators are raised in readily available biological fluids. When taken as a whole, these findings show that LPO is an early feature throughout the progression of the disease and can be regarded as a crucial pathway for targeted therapeutics. It can also be used to improve diagnostic accuracy for early subject detection during the prodromal phase [61,62,63]. The determination of MDA (malondialdehyde) levels is one of the most studied biomarkers for measuring oxidative damage, in this case in lipids [64], and is very useful for measuring the efficacy of a nutritional supplement in the animal’s oxidative stress. To compensate for this ROS-mediated toxicity, cells develop various protective mechanisms. The antioxidant defense system includes many components; a deficiency in any of these components can lead to a general deterioration of antioxidation. TAS determination reflects this overall antioxidation state derived from all molecules with antioxidant power [65,66,67].

Although substantial experimental and post-mortem data confirm the presence of oxidative damage in the brains of individuals with AD, the reliability of peripheral blood markers as indicators of oxidative stress remains debated. A clinical study in AD patients reported elevated levels of 4-hydroxy-2-nonenal (4-HNE), a marker of lipid peroxidation, whereas such changes were not detected when using the more commonly applied marker malondialdehyde (MDA) [68,69]. Although only a non-significant trend was observed for one lipid peroxidation marker (MDA), our results indicate that supplementing the animals’ regular diet with Neurofabine-C may help protect hippocampal tissue from Aβ-induced neurotoxicity, the model of neurodegeneration employed in this study. This effect may be linked to an increase in total antioxidant status (TAS), reflecting the combined action of serum antioxidant molecules. Future studies using additional oxidative markers could provide further support for this hypothesis. Evidence from human research also points to the protective value of antioxidant-related interventions. A study published in PLOS ONE followed 168 older adults with mild cognitive impairment (MCI), a condition characterized by memory and language deficits that exceed normal aging and can precede AD or other dementias. Participants who received high-dose oral vitamins (folic acid [B9], B6, and B12) for two years experienced 30% less brain atrophy compared to the placebo group [70]. Typically, brain volume decreases by about 0.5% annually after the age of 60, with rates doubling in MCI and reaching approximately 2.5% per year in AD. A subsequent randomized controlled trial in elderly individuals at elevated risk for dementia (MCI according to the 2004 Petersen criteria) confirmed these findings. After two years of high-dose vitamin B supplementation (0.8 mg folic acid, 20 mg vitamin B6, 0.5 mg vitamin B12), overall brain shrinkage was significantly reduced. Moreover, in gray matter regions most susceptible to AD pathology, including the medial temporal lobe, vitamin treatment diminished atrophy by as much as sevenfold [71].

β-Amyloid 1-42 (βA42) protein levels were analyzed since they are associated with cerebral amyloidosis and cognitive decline with progression to dementia in patients and are currently considered the most specific marker for Alzheimer’s dementia [72]. Neurons derived from Alzheimer’s patients showed higher levels of extracellular amyloid types βA42 and βA40 [73]. High levels of basal amyloid in the brain were associated with lower simultaneous visual memory and more pronounced declines in language, visuospatial ability, and mental status. Amyloid pathology, like neurodegeneration, has been shown to have detrimental, and in part synergistic, effects on cognitive status. Plasma βA42 concentrations and the βA42/βA40 ratio have been recognized as potential indicators of AD-related pathology in cognitively normal individuals reporting subjective cognitive impairment (SCI) [74]. In one study including participants with SCI, mild cognitive impairment (MCI), and AD dementia and cognitively healthy controls, weak positive correlations were observed between plasma and cerebrospinal fluid (CSF) βA42 and βA40 levels. In contrast, plasma βA42 showed a negative correlation with neocortical amyloid deposition as measured by PET imaging. Plasma βA42 and βA40 levels were reduced in AD dementia compared to all other diagnostic groups. In the preclinical or prodromal phases of AD, including amyloid-positive controls, subjective cognitive decline (SCD), and mild cognitive impairment (MCI), plasma βA42 concentrations show a moderate decrease, while βA40 levels remain stable. Overall, βA levels in plasma are only slightly reduced at the dementia stage, suggesting that significant alterations in βA metabolism appear later in the periphery than in the brain [75]. Drawing on prior findings [72,73,74,75], our results indicate that Neurofabine-C may exert beneficial effects on cognition, consistent with our βA42 data at both the peripheral (serum) and central (neocortical) levels. Furthermore, we confirm the negative association reported in earlier studies [75] between circulating βA42 and amyloid accumulation in the neocortex. Given that (i) βA is a significant constituent of senile plaques, (ii) it plays a central role in the onset and progression of AD, and (iii) numerous in vitro and in vivo studies have shown that βA-induced neuronal and glial damage is mediated by oxidative stress [76], we propose that the reduction in brain βA, and the associated improvement in cognitive function may be linked to the enhanced antioxidant capacity observed in treated mice.

The combined changes in *Dnmt3a* and *Hdac3* expression in the hippocampal regions of Neurofabine-C-treated 3xTg-AD mice point to a chromatin state that supports memory and neural maintenance. DNMT3A is essential for hippocampal-dependent memory formation and activity-regulated gene expression; deletion of *Dnmt3a* in adult forebrain neurons affects long-term potentiation and spatial learning [77,78]. HDAC3 is a negative regulator of memory consolidation, and learning-associated gene expression is normalized, and hippocampus memory formation is increased by genetic deletion or pharmacological suppression [79,80]. The elevation of *Dnmt3a* expression concurrent with the reduction in Hdac3 is therefore consistent with a shift toward increased transcriptional competency for memory-related genes, supported by the parallel increase in global cytosine methylation levels. The adult brain contains high levels of 5-hydroxymethylcytosine (5hmC), which is particularly enriched in neuronal gene bodies and genes associated with synaptic function [81]. Many colorimetric assays, however, quantify total modified cytosines without distinguishing between 5mC and 5hmC [82,83]. Moreover, bulk tissue measurements reflect contributions from multiple cell types with distinct epigenetic profiles [84]. Given the parallel changes in methyltransferase and deacetylase expression, the elevated global methylation signal suggests increased capacity for activity-dependent transcriptional regulation in hippocampal neurons. Nonetheless, future studies using locus-specific and cell-type-resolved approaches will be necessary to characterize these changes fully.

The increase in the genes *Atm* and *Tp73*, which promote DNA repair and neuronal survival, demonstrated the coordinated overexpression of the DNA damage response system. ATM loss makes neurons more susceptible to degeneration, as seen in ataxia-telangiectasia [85,86]. ATM kinase synchronizes genome maintenance with activity-dependent transcriptional programs in neurons. Hippocampal development, adult neurogenesis, and post-mitotic neuronal survival are all supported by TP73, a member of the p53 family, and these processes are compromised by loss-of-function mutations [87,88]. The substantial increases in both *Atm* and *Tp73* expression following treatment are consistent with enhanced DNA repair capacity and neuronal survival signaling in the hippocampus. *FOXO3* upregulation in cocoa-treated 3xTg-AD mice suggests activation of neuroprotective stress-response pathways in the hippocampus, including antioxidant defense, DNA repair, and autophagy [89,90]. This upregulation aligns with prior evidence implicating FOXO3 in promoting longevity and protecting against AD pathology in animal models and human studies [91,92].

The decrease in *HIF-1α* expression in Neurofabine-C-treated 3xTg-AD mice provides a mechanistic link between the epigenetic changes and the reduced neocortical Aβ1-42 levels we measured. HIF-1α promotes amyloidogenic processing through transcriptional upregulation of BACE1 and activation of the γ-secretase complex [93,94]. Thus, the downregulation of *HIF-1α* transcripts is consistent with the observed reduction in amyloid burden, although we acknowledge that HIF-1α regulation occurs primarily at the protein level through post-translational modifications.

The observed reduction in *Apoe* expression in Neurofabine-C-treated 3xTg-AD mice may indicate reduced glial activation secondary to the lower amyloid burden. In the adult brain, APOE is primarily produced by astrocytes, with context-dependent expression in activated microglia. Both cell types influence amyloid plaque dynamics and neuroinflammatory responses [95,96,97]. Reduced *Apoe* mRNA expression in Neurofabine-C-treated 3xTg-AD mice is consistent with a less reactive glial environment, which accompanies the reduced cortical Aβ levels. These molecular changes may be attributed to the polyphenol-rich intervention. Cocoa flavanols, particularly (-)-epicatechin, improve hippocampus-dependent memory in rodents and dentate gyrus function and cerebral blood flow in humans [98,99]. Polyphenolic compounds also modulate epigenetic enzymes, including DNMTs and HDACs, and influence redox and inflammatory pathways that regulate hippocampal gene expression [100,101]. The bioactive compounds in our Neurofabine-C extract, therefore, provide a plausible basis for the coordinated changes in chromatin-modifying enzymes and stress-response pathways observed in this study.

Collectively, these epigenetic and transcriptional changes suggest that dietary intervention with Neurofabine-C promotes a neuroprotective profile in 3xTg-AD mice, characterized by increased DNA methylation, reduced histone deacetylation, strengthened DNA repair, and decreased *Apoe* expression associated with glial reactivity. This signature is consistent with improvements in antioxidant capacity and reduced brain amyloid burden observed in the same animals. Our data support a model in which Neurofabaline-C-derived polyphenols foster an environment favorable to neuronal survival by enhancing DNA repair, promoting chromatin remodeling, and reducing amyloidogenic signaling in the hippocampus.

Finally, the immunohistochemical results obtained from AD transgenic mice brain sections, have shown that Neurofabaline-C treatment induces a neuroprotective effect against neuro-degeneration hallmarks. Based on the quantification data of specific antibody markers, the present treatment shows a protective effect in the cortical and hippocampal layers of the brain, exhibiting a double-fold density of immunoreactive Pax6 cells, when compared with control mice. This increase pattern of PAX6 neurons demonstrates the regulatory effect of treatment against neuropathological agents that inhibit the neurodifferentiation in these cortical regions. In transgenic mouse models of AD, treatment resulted in a dense distribution of NeuN-immunoreactive cells, particularly concentrated in the outer layers of the entorhinal cortex, suggesting a neuroprotective effect on cortical neuronal development. These findings are consistent with previous reports in other murine models [102,103,104]. Clinical evidence also supports the role of AtreMorine in modulating dopamine neurotransmission, reducing inflammation, and promoting neuroprotection. In patients with PD, administration of AtreMorine, a fava-bean-derived extract, has been shown to increase plasma dopamine concentrations 200- to 500-fold [26,27], an effect attributed to its high natural L-DOPA content (approximately 20 mg/g). When combined with standard antiparkinsonian therapies, AtreMorine permits a 25–50% reduction in the dosage of conventional drugs, providing direct benefits for managing core PD symptoms while lowering the incidence of both short- and long-term adverse effects [26,27].

Summing up, the two extracts from the hybrid compound Neurofabine-C seem to possess antioxidant and antidegenerative properties, and the observed effect against Alzheimer’s disease pathology in mice may be due to the neuroprotective bioactive protein cocktail present in the synergic extract. The present formulation of a novel nutritional supplement based on biofunctional food ingredients is highly promising against the pathological hallmarks of degenerative diseases.

## 5. Conclusions

Plant-based dietary patterns are a feasible and helpful way to prevent chronic diseases, as evidenced by preclinical and clinical research conducted over the past ten years. Understanding the processes of natural product mixes that exhibit a positive effect against degenerative diseases was the goal of the current study. Our research has concentrated on the potential anti-inflammatory properties of Neurofabine-C, a hybrid compound with bioactive phytoconstituents of *Vicia faba* bean and cocoa. Here, we have shown that by altering neuroinflammation biomarkers in mice, Neurofabine-C may be able to prevent cognitive deterioration. The current findings are novel as they indicate that cocoa may affect, either directly or indirectly, signal transduction pathways associated with brain inflammation and neurodegenerative indicators. This suggests that cocoa extract functions as a neuroprotective agent by safeguarding against neurological disorders associated with oxidative stress. Furthermore, by disrupting redox-regulated pathways, cacao’s antioxidant constituents can diminish the degradation of tryptophan, the synthesis of serotonin, and the levels of inflammatory indicators. So, the combination of cocoa and faba bean extracts in Neurofabine-C works together to raise dopamine levels in people with PD. This has an effect of protecting neurons in the SNpc by slowing down the breakdown of dopamine. In conclusion, this study aims to provide some insight into the long-term search for efficient treatment and prevention strategies for neurodegenerative disorders, which have a significant influence on public health and the quality of life for a large number of individuals globally.

## Figures and Tables

**Figure 1 genes-16-01214-f001:**
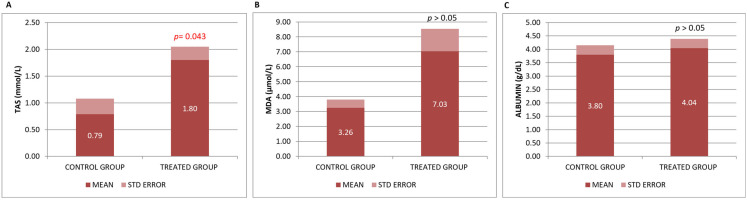
(**A**) Total antioxidant status (TAS), (**B**) MDA and (**C**) albumin levels in serum. Data are shown as mean ± SEM. Statistical significance was determined by Student’s *t*-test: * *p* < 0.05.

**Figure 2 genes-16-01214-f002:**
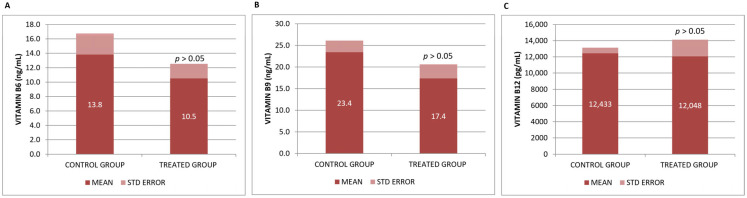
(**A**) Vitamin B6, (**B**) Vitamin B9, and (**C**) Vitamin serum concentrations. Data are shown as mean ± SEM. Statistical significance was determined by Student’s *t*-test: * *p* < 0.05.

**Figure 3 genes-16-01214-f003:**
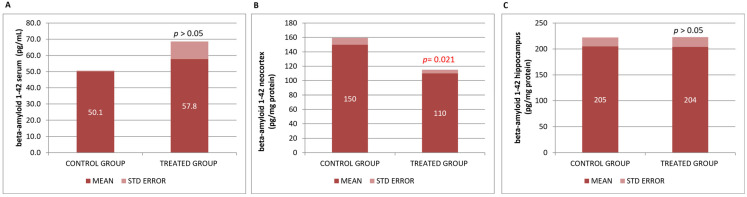
(**A**) βA42 in blood serum, (**B**) neocortex βA42, and (**C**) hippocampus βA42 concentrations. Data are shown as mean ± SEM. Statistical significance was determined using Student’s *t*-test: * *p* < 0.05.

**Figure 4 genes-16-01214-f004:**
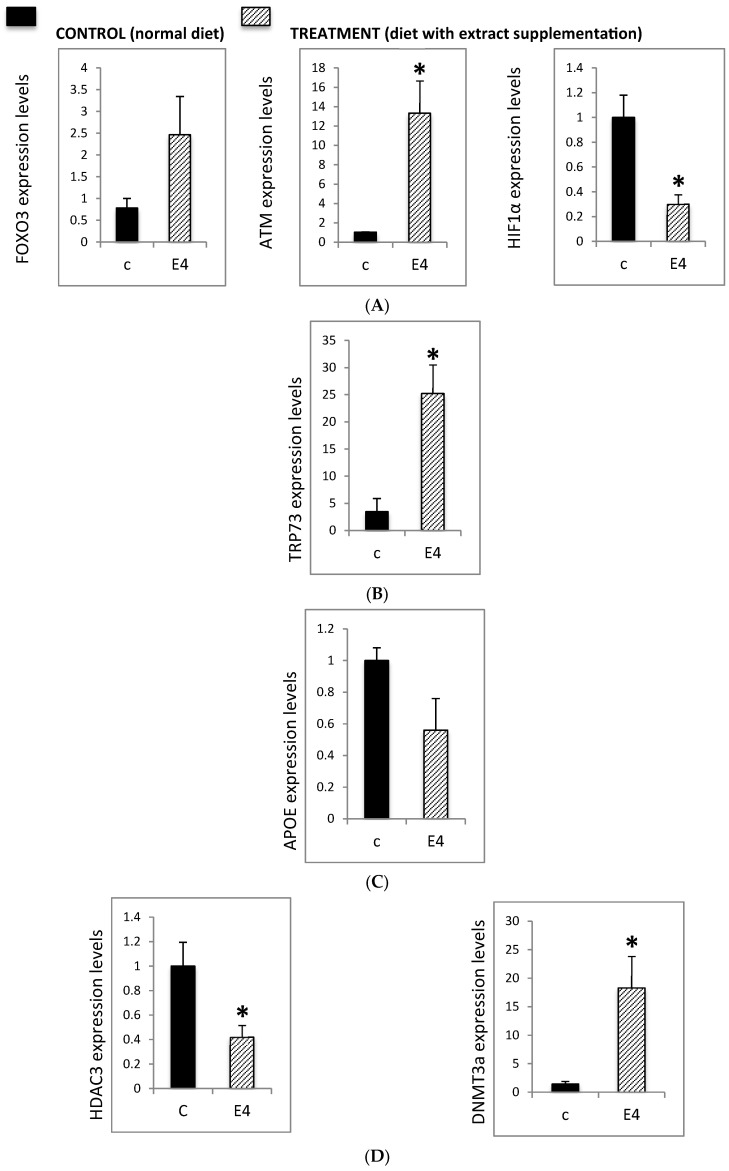
(**A**) Expression levels of *FOXO3*, *ATM*, and *HIF1α* in hippocampal tissue from 3xTg-AD mice. Mice were fed either a regular diet (C, control, black bars) or a regular diet supplemented with Neurofabine-C (E4, treatment, gray bars) for five weeks. Gene expression was analyzed by qRT-PCR and normalized to a housekeeping gene, expressed as fold change relative to control. Data are shown as mean ± SEM. Statistical significance was determined using Student’s *t*-test: * *p* < 0.05. *n* = 3–5 per group. (**B**) *TRP73* expression levels in hippocampal tissue from 3xTg-AD mice. Mice were fed either a regular diet (C, control, black bars) or a regular diet supplemented with Neurofabine-C extract (E4, treatment, gray bars) for five weeks. Gene expression was analyzed by qRT-PCR and normalized to a housekeeping gene, expressed as fold change relative to control. Data represent mean ± SEM. Statistical significance was determined using Student’s *t*-test: * *p* < 0.05. *n* = 3–5 per group. (**C**) *APOE* expression levels in hippocampal tissue from 3xTg-AD mice. Mice were fed either a regular diet (C, control, black bars) or a regular diet supplemented with Neurofabine-C (E4, treatment, gray bars) for five weeks. Gene expression was analyzed through qRT-PCR and normalized to housekeeping genes, expressed as fold change relative to control. *APOE* expression decreased by approximately 40% following treatment. Data represent mean ± SEM. Statistical significance was determined using Student’s *t*-test. *n* = 3–5 per group. (**D**) Expression levels of epigenetic regulatory enzymes *DNMT3A* and *HDAC3* in hippocampal tissue from 3xTg-AD mice. Mice were fed either regular diet (C, control, black bars) or a regular diet supplemented with Neurofabine-C (E4, treatment, gray bars) for five weeks. Gene expression was analyzed through qRT-PCR and normalized to a housekeeping gene, expressed as fold change relative to control. Data represent mean ± SEM. Statistical significance was determined using Student’s *t*-test: * *p* < 0.05. *n* = 3–5 per group. (**E**) Global 5-methylcytosine (5mC) levels in hippocampal tissue from 3xTg-AD mice. Mice were fed either a regular diet (C, control, black bars) or a regular diet supplemented with Neurofabine-C (E4, treatment, gray bars) for five weeks. Global DNA methylation was quantified using a colorimetric assay and expressed as 5mC content per hippocampus. Data are presented as mean ± SEM. Statistical significance determined using Student’s *t*-test: * *p* < 0.05. *n* = 3–5 per group.

**Figure 5 genes-16-01214-f005:**
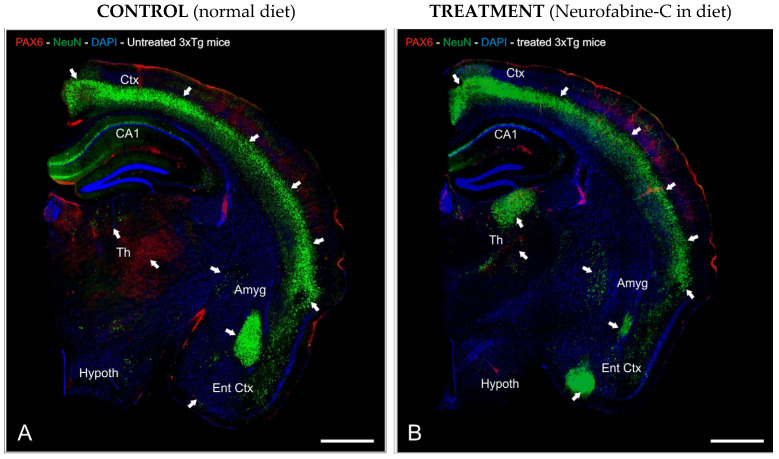
Neurofabine-C protective effect against neurodegeneration in AD transgenic mice. (**A**,**B**): Half-brain segment sagittal slices immunostained with anti-NeuN (green) and anti-Pax6 (red) antibodies. DAPI (blue) was used as a counterstain for the nuclei. Images were prepared for projection with the highest intensity possible. The areas of the brain with the highest levels of immunoreactivity are indicated by the white arrows. Scale bar: 100 μm. Abbreviations: Amyg, Amygdala; CA1, hippocampal layer; Ctx, Cortex; DAPI, neuronal nuclei marker; DG, Dentate Gyrus; Ent Ctx, Entorhinal Cortex; Hypoth, Hypothalamus; Th, Thalamus.

**Figure 6 genes-16-01214-f006:**
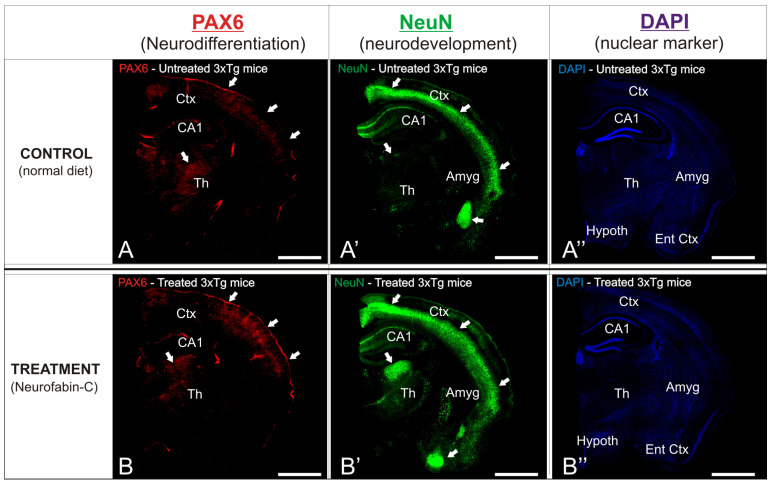
(**A**,**B**): Antibodies against Pax6 (red, neurodifferentiation marker) and NeuN (green, neurodevelopmental marker) were used to immunostain sagittal brain slices. DAPI (blue) was used as a counterstain for the nuclei. Images were manipulated to project maximum intensity. The brain’s most immunoreactive areas are indicated by the white arrows. Treated mice were given Neurofabine-C supplementation in the diet. (**A’**,**B’**): NeuN immunostaining (green) showing neuronal distribution and density in untreated (**A’**) and Neurofabine-C–treated (**B’**) 3xTg mice. (**A”**,**B”**): DAPI nuclear counterstaining (blue) illustrating overall cellular distribution in untreated (**A”**) and treated (**B”**) 3xTg mice. Scale bars: 100 μm.

**Table 1 genes-16-01214-t001:** The diets/treatment regime.

	Weeks of Treatment				
	1	2	3	4	5
Gr A (Regular Diet)	RD	RD	RD	RD	RD
Gr B (Neurofabine-C + Regular Diet)	RD + CES	RD + CES	RD + CES	RD + CES	RD + CES

**Table 2 genes-16-01214-t002:** Biochemical changes observed in mice treated with Neurofabine-C.

PARAMETER	UNITS	GROUP	N	MEAN	STANDARD DEVIATION	STANDARD ERROR	*T*-TEST Sig (*p*)
IMMUNE EFFECT/ENERGETIC EFFECT/ANTIANEMIC EFFECT							
**LEUCOCYTES**	×10^9^/L	CONTROL	3	2.0767	0.76422	0.44122	
		TREATMENT	5	16,160	0.85949	0.38438	*p* > 0.05
**RED BLOOD CELLS**	×10^12^/L	CONTROL	3	9.5533	0.83722	0.48337	
		TREATMENT	5	9.6320	0.32729	0.14637	*p* > 0.05
**HEMOGLOBIN**	g/dL	CONTROL	3	14.4333	1.34288	0.77531	
		TREATMENT	5	14.3600	0.54589	0.24413	*p* > 0.05
**HEMATOCRIT**	%	CONTROL	3	50.1000	4.66047	2.69072	
		TREATMENT	5	49.4400	2.46739	1.10345	*p* > 0.05
**MCV**	fL	CONTROL	3	52.4333	0.68069	0.39299	
		TREATMENT	5	51.3200	1.65136	0.73851	*p* > 0.05
**MCH**	pg	CONTROL	3	15.1000	0.34641	0.20000	
		TREATMENT	5	14.9200	0.24900	0.11136	*p* > 0.05
**MCHC**	g/dL	CONTROL	3	28.8000	0.26458	0.15275	
		TREATMENT	5	29.0800	0.40866	0.18276	*p* > 0.05
**PLATELETS**	×10^9^/L	CONTROL	3	871.0000	58.94913	34.03430	
		TREATMENT	5	932.6000	189.22685	84.62482	*p* > 0.05
**MPV**	fL	CONTROL	3	5.4000	0.10000	0.05774	
		TREATMENT	5	5.6600	0.08944	0.04000	*p* > 0.05
**PCT**	%	CONTROL	3	0.4700	0.02700	0.01559	
		TREATMENT	5	0.5290	0.11495	0.05141	*p* > 0.05
NUTRITIONAL STATUS							
**VITAMIN B6**	ng/mL	CONTROL	3	13.8250	5.07250	2.92861	
		TREATMENT	5	10.5234	4.48124	2.00407	*p* > 0.05
**VITAMIN B9**	ng/mL	CONTROL	3	23.4408	4.61265	2.66311	
		TREATMENT	5	17.3630	7.30684	3.26772	*p* > 0.05
**VITAMIN B12**	pg/mL	CONTROL	2	12,433.12	1004.97551	710.62500	
		TREATMENT	5	12,047.75	4632.17184	2071.57022	*p* > 0.05
**ALB**	g/dL	CONTROL	3	3.8000	0.60828	0.35119	
		TREATMENT	5	4.0400	0.77974	0.34871	*p* > 0.05
ANTIOXIDANT EFFECT							
**TAS SERUM**	mmol/L	CONTROL	3	0.7867	0.50362	0.29077	
		TREATMENT	5	1.7960	0.55851	0.24977	*p* = 0.043
**MDA PLASMA**	µmol/L	CONTROL	3	3.2633	0.94214	0.54395	
		TREATMENT	5	7.0280	3.26481	1.46007	*p* > 0.05
COGNITIVE STATUS (ALZHEIMER BIOMARKER)							
**βA 1-42 SERUM**	pg/mL	CONTROL	3	50.100	0.90067	0.52000	
		TREATMENT	5	57.756	24.44359	10.93151	*p* > 0.05
**βA 1-42 NEOCORTEX**	pg/mg protein	CONTROL	3	150.166	16.26012	9.38779	
		TREATMENT	3	110.510	9.06759	5.23517	*p* = 0.021
**βA 1-42 HIPPOCAMPUS**	pg/mg protein	CONTROL	3	205.253	30.19689	17.43418	
		TREATMENT	3	203.680	33.23732	19.18958	*p* > 0.05

## Data Availability

The original contributions presented in this study are included in the article/Appendix A. Further inquiries can be directed to the corresponding author.

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
