# Peer review of "Epigenetic Modulation and Neuroprotective Effects of Neurofabine-C in a Transgenic Model of Alzheimer’s Disease"

_genes, 2025, doi:10.3390/genes16101214_

Round 1

Reviewer 1 Report

Comments and Suggestions for Authors

In this manuscript, the authors presented the results from immunohistochemical and biochemical analysis of the effects of Neurofabine-C treatment on the triple-transgenic 3xTg-AD mice. They found that TAS significantly increased and βA 1-42 in the neocortex significantly decreased in the treatment group. They reported up-regulation of neuroprotective genes and down-regulation of HIF1α and APOE. They also found coordinated increases of DNMT3A and decreases of HDAC3, accompanied by the increase of global 5-methylcytosine levels. Although the idea of Neurofabine-C treatment is interesting and the results might be of interest to the readers, several issues need to be addressed.

Major issues:

  1. It's concerning that the authors didn't perform all these measurements in control mice with a regular diet and Neurofabine-C treatment. Without these two groups of control mice, it's impossible to know if Neurofabine-C actually rescues/reverses the phenotypes in the 3xTG-AD mice.
  2. The number of mice per group reported throughout the manuscript is not consistent. In the method section (lines 157-158) the authors said "five mice were used in the control group A (without treatment in food) and 10 in treatment group B (with Neurofabine-C enriched food)". But they reported "N control = 3 and N treatment = 5" in Table 2 and "n = 3–5 per group" in Figure 2.
  3. All the data are shown in bar plots in Figure 1 and Figure 2. It's recommended nowadays to show individual data points along with boxplot/violin plots to better visualize the distribution of the data. As pointed out earlier in #2, there is no way to confirm how many mice were actually included in each group. Table 2 only provides the summarized statistics for each group, and there are no supplementary table that reports the raw data points for each of these metrics.
  4. In Figure 2E, what are the units of the global 5-methylcytosine levels? Were those derived from a standard curve from the colorimetric assay? It's difficult to understand the meaning of a value of 0.005 or 0.03 without context. Do they mean lowly methylated or highly methylated? Some sort of positive and negative controls should be included.

Minor issues:

  1. The figure legend for Figure 1 is not self-explanatory. What do the dark and light red proportions of each bar mean? What are the numbers labelled on the bars? What statistical test was used to get the p-values? All these should be described clearly in the figure legend.
  2. In the column name of Table 2, "Media" seems to be a typo. Is it supposed to be "mean" or "median"?
  3. The texts and the asterisks in Figure 2A are offset.
  4. Some typos: in line 16 "The faba bean extract (Vicia faba) extract obtained..." should be ". The faba bean (Vicia faba) extract obtained..."; in line 150 "...and backcross-bed in our laboratory" should be "and backcrossed in our laboratory"? 

Author Response

The points raised by the reviewers have been addressed as follows (reviewers’ comments are italicized); “Changes were highlighted” to reflect modifications to the revised manuscript.

REVIEWER 1

Comment 1:

It's concerning that the authors didn't perform all these measurements in control mice with a regular diet and Neurofabine-C treatment. Without these two groups of control mice, it's impossible to know if Neurofabine-C actually rescues/reverses the phenotypes in the 3xTG-AD mice.

Response:

We appreciate this comment by the reviewer, and agree that a complete study would have to perform all de possible factors, variables and measurements that may influence the final results, however, our major goal in this first approach was to simplify the experimental design and emphasise the difference effect on transgenic mice treated with or without Neurofabine-C. We totally agree with your concern, and we will extend the number of groups as you propose, in the next research study, taking the present one as a preliminary data.    

REVIEWER 1

Comment 2:

The number of mice per group reported throughout the manuscript is not consistent. In the method section (lines 157-158) the authors said "five mice were used in the control group A (without treatment in food) and 10 in treatment group B (with Neurofabine-C enriched food)". But they reported "N control = 3 and N treatment = 5" in Table 2 and "n = 3–5 per group" in Figure 2.

Response:

We thank the reviewer for this comment and acknowledge that the rationale for the sample sizes was not clearly described in the original manuscript.

Five mice were included in the control group and ten in the treatment group. All samples were processed for biochemical assays. In some cases, however, technical limitations prevented reliable quantification, such as insufficient sample material, hemolysis or degradation during handling, assay signals falling below the detection limit of the kit, or intra-assay variability exceeding acceptable thresholds. Data from such samples were therefore not available for statistical analysis. As a result, the number of valid measurements was three in the control group and five in the treatment group for most parameters, two in the control group and five in the treatment group for vitamin B12, and three per group for β-amyloid measurements in brain tissue, as detailed in the text.

REVIEWER 1

Comment 3:

All the data are shown in bar plots in Figure 1 and Figure 2. It's recommended nowadays to show individual data points along with boxplot/violin plots to better visualize the distribution of the data. As pointed out earlier in #2, there is no way to confirm how many mice were actually included in each group. Table 2 only provides the summarized statistics for each group, and there are no supplementary table that reports the raw data points for each of these metrics.

Response:

We appreciate this comment by the reviewer, and we have included the biochemical raw data in supplementary data section to solvent this comment according to your recommendations.

REVIEWER 1

Comment 4:

In Figure 2E, what are the units of the global 5-methylcytosine levels? Were those derived from a standard curve from the colorimetric assay? It's difficult to understand the meaning of a value of 0.005 or 0.03 without context. Do they mean lowly methylated or highly methylated? Some sort of positive and negative controls should be included.

Response:

We appreciate this comment by the reviewer, and we agree that the explanation of the methodology used for the calculation is unclear. We have followed the manufacturer's instructions for relative quantification. To do this, we have used the following formula:

5-mC= ((Sample OD-Negative OD)/ DNA (ng))/((Positive OD-Negative OD)x2/amount positive control (ng))

We have modified the description in the material and methods description.

REVIEWER 1

Comment 5:

The figure legend for Figure 1 is not self-explanatory. What do the dark and light red proportions of each bar mean? What are the numbers labelled on the bars? What statistical test was used to get the p-values? All these should be described clearly in the figure legend.

Response:

We appreciate this comment by the reviewer, and we have modified the figures and legends as recommended.

REVIEWER 1

Comment 6:

In the column name of Table 2, "Media" seems to be a typo. Is it supposed to be "mean" or "median"?

Response:

We appreciate this comment by the reviewer, and we change the word “media” to “mean”.

REVIEWER 1

Comment 6:

The texts and the asterisks in Figure 2A are offset.

Response:

We appreciate this comment by the reviewer, and we have corrected those figures.

REVIEWER 1

Comment 6:

Some typos: in line 16 "The faba bean extract (Vicia faba) extract obtained..." should be ". The faba bean (Vicia faba) extract obtained..."; in line 150 "...and backcross-bed in our laboratory" should be "and backcrossed in our laboratory"?

Response:

We appreciate these corrections by the reviewer. Changes in text have been made accordingly.

Reviewer 2 Report

Comments and Suggestions for Authors

Hello Authors, 

This is an interesting study assessing the effects of Neurofabine-C on a AD mouse model. I have some comments: 

  1. Introduction. First, third, and seven paragraphs are missing references.
  2. Material and Methods. There is no description of how you extracted the DNA for the methylation analysis. 
  3. Results:
    1. Please, define TAS.
    2. Please, refer to Figure 1A, B, C on the text and also, create a panel with A, B, C. If you put these figures independently, they should name it as Figure 2, 3,...
    3. Define ATM. Same as above with Figure 2A, B,... In 3.2.1., the introductory paragraph should be moved to Introduction/discussion. This section is for results. 
    4. Discussion. Many of the references included on the discussion are mostly facts. These could be included on the introduction. I suggest focusing the discussion on relating the findings of this research with previous similar studies. 
  4. Conclusion. Please, check, neurofabine is misspealling. 

Thank you very much.

Author Response

The points raised by the reviewers have been addressed as follows (reviewers’ comments are italicized); “Changes were highlighted” to reflect modifications to the revised manuscript.

REVIEWER 2

Comment 1:

Introduction. First, third, and seven paragraphs are missing references.

Response:

We appreciate this comment by the reviewer, and agree that it is important to address the statements with references. We have included references in all paragraphs as requested.

REVIEWER 2

Comment 2:

Material and Methods. There is no description of how you extracted the DNA for the methylation analysis.

Response:

We appreciate this comment by the reviewer, and we regret not having included it. This is the text we have added to the materials and methods section of the article: “DNA was extracted from mice hippocampi with the Qiagen DNA Mini Kit (Qiagen, Hilden, Germany) following the instructions of the commercial establishment, and only DNA samples with 260/280 and 260/230 ratios greater than 1.8. were used.”

REVIEWER 2

Comment 3:

Results:Please, define TAS.

Response:

We appreciate this comment by the reviewer, and we include TAS definition in the text (line 303) and reference.

REVIEWER 2

Comment 4:

Results: Please, refer to Figure 1A, B, C on the text and also, create a panel with A, B, C. If you put these figures independently, they should name it as Figure 2, 3,...

Response:

We appreciate this comment by the reviewer, and we have modified the distribution of figures creating panels and referred to them on the text as you recommended.

REVIEWER 2

Comment 5:

Results: Define ATM. Same as above with Figure 2A, B,... In 3.2.1., the introductory paragraph should be moved to Introduction/discussion. This section is for results.

Response:

We appreciate this comment by the reviewer, and changes have been made in the text, accordingly.

REVIEWER 2

Comment 6:

Discussion. Many of the references included on the discussion are mostly facts. These could be included on the introduction. I suggest focusing the discussion on relating the findings of this research with previous similar studies.

Response:

We appreciate this comment by the reviewer, and we have rewritten some paragraphs of the discussion as suggested.

REVIEWER 2

Comment 7:

Conclusion. Please, check, neurofabine is misspealling.

Response:

We agree with this comment and we corrected the words accordingly.

Reviewer 3 Report

Comments and Suggestions for Authors
  1. In Line 30, please add a space in the “fourfold”.
  2. In Figure 1A-1I, authors used two different colors in each column, what do the dark and light red mean in this figure? Please add the clarification either in the figure or in the figure legend. If the p>0.05, authors can put the figures into supplementary material and represent the significant ones in the main text. Additionally, please merge the sub-figures as one.
  3. Figure 2A needs to be adjusted to make the stars and the words at the right place. Authors can merge sub-figures 2A-2E as one figure and label as sub-titles.
  4. In table 2, columns like the standard error show number “,44122”, what does it mean? Please revise these numbers as correct format.

Author Response

The points raised by the reviewers have been addressed as follows (reviewers’ comments are italicized); “Changes were highlighted” to reflect modifications to the revised manuscript.

REVIEWER 3

Comment 1:

In Line 30, please add a space in the “fourfold”.

Response:

We totally agree with this comment and we corrected the word accordingly.

REVIEWER 3

Comment 2:

In Figure 1A-1I, authors used two different colors in each column, what do the dark and light red mean in this figure? Please add the clarification either in the figure or in the figure legend. If the p>0.05, authors can put the figures into supplementary material and represent the significant ones in the main text. Additionally, please merge the sub-figures as one.

Response:

We appreciate this comment by the reviewer, and agree the organization of the figures is improvable. Following your indications and comments of reviewer 2, we modified the distribution of figures in three panels with different numbers and included the legends in each one.

REVIEWER 3

Comment 3:

Figure 2A needs to be adjusted to make the stars and the words at the right place. Authors can merge sub-figures 2A-2E as one figure and label as sub-titles.

Response:

We appreciate this comment by the reviewer, and we have modified the distribution of figures creating panels and referred to them on the text as you recommended. Stars and words have been moved to the right place.

REVIEWER 3

Comment 4:

In table 2, columns like the standard error show number “,44122”, what does it mean? Please revise these numbers as correct format.

Response:

We appreciate and I agree with this comment. We changed the number format as you recommended.

Round 2

Reviewer 1 Report

Comments and Suggestions for Authors

The authors have addressed most of the issues, except that the text and asterisks in the new Figure 4A are offset. I have no more comments to add.

Author Response

No reply needed. 

Reviewer 2 Report

Comments and Suggestions for Authors

Dear Authors, 

Thank you very much for addressing my comments! The manuscript has improved a lot. 

Thank you!

Author Response

No reply needed